# Association between Social Trust and Metabolic Syndrome in a Previously Healthy Population—A Longitudinal Cohort Study in South Korea

**DOI:** 10.3390/ijerph17165629

**Published:** 2020-08-05

**Authors:** Hyeah Park, Seulggie Choi, Kyae Hyung Kim, EunKyo Kang, Ahryoung Ko, Sang Min Park

**Affiliations:** 1Department of Family Medicine, Seoul National University Hospital, Seoul 03080, Korea; hyeahp@snu.ac.kr (H.P.); truwhat@gmail.com (K.H.K.); ekherb@naver.com (E.K.); 2Department of Biomedical Sciences, Seoul National University Graduate School, Seoul 03080, Korea; seulggie@gmail.com; 3Institute for Public Health & Medical Service, Seoul National University Hospital, Seoul 03080, Korea; ahryoungko@gmail.com

**Keywords:** social trust, metabolic syndrome

## Abstract

*Background:* Social trust, assessed by the trustworthiness of one another in a community, is known to have beneficial effects on health outcomes. However, the impact of social trust on metabolic syndrome (MetS) is unclear. *Methods:* The study subjects were extracted from the Korean National Health Insurance Service, and social trust was obtained from the Korean Community Health Survey (KCHS). Previously healthy participants were followed up from 1 January 2010 to 31 December 2011, and again from 1 January 2012 to 31 December 2013 for waist circumference, blood pressure, fasting blood glucose, triglycerides and high-density lipoprotein cholesterol (HDL-C). Multivariate logistic regression was used to calculate the adjusted odds ratios (aORs) with 95% confidence intervals (CIs) for newly developed MetS according to social trust quintiles. Stratified analyses were performed to determine the relationship between lifestyle behaviors and social trust. *Results:* Compared to the participants within the first quintile of social trust, those in the remaining quintiles had lower risks of developing MetS. The aOR with the 95% CI was 0.88 (0.79–0.98) in the 5th quintile group of social trust. Among the diagnostic criteria for MetS, waist circumference and HDL-C were statistically significant with aORs of 0.91 (0.84–0.99) and 0.88 (0.80–0.95) in the 5th quintile group. The stratified analyses showed protective effects of positive lifestyle behaviors. The aORs with 95% CIs were 0.85 (0.74–0.99) in never smokers, 0.82 (0.70–0.95) in non-drinkers and 0.87 (0.76–1.00) in the physically active in the highest level of social trust. *Conclusions*: Higher social trust was associated with a lower incidence of MetS. Therefore, building community with psychosocial support may be helpful in improving public health.

## 1. Introduction

Social capital has been considered an important factor in determining health status since its introduction in the 1990s [1]. Although many dimensions of social capital exist, it has been generally accepted as an asset for promoting beneficial economic, social and health outcomes [2,3]. Among the components of social capital, social trust, as a cognitive component, has been known to facilitate social interaction and the exchange of information [4]. Social trust is usually assessed by the question, “would you say that people can be trusted?” This question evaluates the trustworthiness of one’s neighborhood, which may impact one’s behavior in the community [5]. Because social trust can also influence health behavior, one’s metabolic profiles may change depending on the level of social trust. While abundant investigations have been performed to study the association between social capital and various health outcomes, such as all-cause mortality [6] and depression [7], there is a dearth of information on the relationship between social capital and metabolic syndrome (MetS). With its increasing incidence and predictive value for cardiovascular risks and diabetes [8], MetS has become an important global health issue. Therefore, determining the effects of social trust on MetS may be crucial for public health implications.

Evidence for a significant association between social capital and MetS is lacking and only studies that used proxy measures for MetS are available. While some studies showed a positive correlation between social trust and cardiovascular diseases [9] and obesity [10], others showed null or negative results [11]. A study in Australia found a higher objective crime rate associated with a higher risk of MetS in men, and a higher perceived crime rate associated with a higher risk in women. Both total and violent crime rates were associated with MetS in men, but only the perceived crime rates of neighborhoods were associated with MetS in women [12]. Another study in Canada found a negative correlation between network social capital and waist circumference. It did not, however, find a significant association between social trust and waist circumference [13]. Because the association between MetS and social trust has not yet been established, elucidating the relationship may help improve public health outcomes. As MetS has become such a syndemic [14], the primary prevention of MetS requires more societal and structural changes.

Therefore, the objective of this study was to elicit the association between community-level social trust and MetS using a national cohort study of Korean population data. It was hypothesized that a higher level of social trust was related to a lower likelihood of developing MetS.

## 2. Methods

### 2.1. Study Population

This was a longitudinal, retrospective study that collected health information from existing data. The study population was extracted from the Korean National Health Insurance Service–National Sample Cohort (NHIS-NSC). In South Korea, the NHIS is a universal healthcare system for all Korean citizens, collecting health service utilization records for insurance claim purposes that include outpatient and inpatient hospital visits, health screening examinations, diagnostic and treatment-related procedures and pharmaceutical prescriptions. The health screening exams contain a self-reported questionnaire on lifestyle behaviors, anthropometric measurements and urine and blood tests biannually for enrollees aged 40 years or older. Parts of these data are available for research purposes, and many large-scale epidemiological studies have used the NHIS database. The validity of the database is described in detail elsewhere [15,16].

Among 211,125 participants aged 40 years or older in the NHIS-NSC, 724 enrollees who died before the index date were excluded. Two thousand seven hundred and seven and 836 were excluded for missing values for covariates and MetS criteria, respectively. Those who did not answer social capital-related questions (2069 people) and who were already taking statins, hypertension medication or diabetes medication (116,000 people) were removed from the study. This study only included participants who lived in the metropolitan area. A total of 47,289 participants were excluded from the study for living in rural areas. The participants meeting the inclusion criteria were followed from 1 January 2010 to 31 December 2011, and again from 1 January 2012 to 31 December 2013 for waist circumference, blood pressure, fasting blood glucose, triglycerides and high-density lipoprotein cholesterol (HDL-C).

The study was approved by the Seoul National University Hospital Institutional Review Board (IRB number: E-1806-076-951). Because the NHIS-NSC database is anonymized according to strict confidentiality guidelines prior to distribution, the requirement for informed consent was waived.

### 2.2. Key Variables

Social trust values were measured using the Kawachi method and the details and validity of it have been described in previous papers [17]. The Korean Community Health Survey (KCHS) had a question to assess social trust, which was conducted by the Korean Centers for Disease Control and Prevention in 2011. It is a nationally and district-representative community-based cross-sectional survey that contains community-level information according to administrative district sites [18]. Social trust was assessed by the statement, “the people in my neighborhood can trust one another,” and the responses were categorized into two answers: trusting and non-trusting. Social trust was calculated by determining the proportion of those who answered “yes” to the social trust question for each administrative district site. A total of 253 district sites, with a mean (standard deviation) land area of 55.1 (79.9) km^2^, covers the entire South Korea land mass. The social trust values were then merged with NHIS-NSC according to each participant’s residential district (a total of 253 districts). Rural areas were then excluded and only residents from 74 districts were included in the study. Participants were then categorized into five groups evenly according to the level of social trust, the 1st quintile having the lowest level of social trust and the 5th quintile having the highest level of social trust.

All Korean citizens have universal healthcare access managed by the NHIS, which covers nearly all health care services and biannual health screenings for people 40 years and older. The healthcare database contains waist circumference, fasting blood glucose, HDL-C, and triglyceride levels and blood pressure.

The definition of MetS was derived from revised the National Cholesterol Education Program Adult Treatment Panel III (NCEP ATP III) criteria [19]. It requires at least three of the following components: (1) abdominal obesity (waist circumference ≥90 cm for men, or ≥85 cm for women); (2) triglycerides ≥150 mg/dL and/or drug treatment for elevated triglycerides; (3) HDL-C < 40 mg/dL for men or <50 mg/dL for women; (4) systolic blood pressure ≥130/85 mmHg or antihypertensive medication treatment and/or a history of hypertension; and (5) FSG ≥ 100 mg/dL and/or treatment with medications for type II diabetes mellitus.

### 2.3. Statistical Analysis

Multivariate logistic regression was used to determine the adjusted odds ratios (aORs) with 95% confidence intervals (CIs) for MetS as a composite outcome and each component of MetS (waist circumference, fasting blood glucose, HDL-C, triglyceride levels and blood pressure). The incidence of developing MetS was calculated compared to the 1st quintile of the social trust group. Social trust was divided into five groups, with the lowest being the 1st quintile and the highest being the 5th quintile. The covariates considered included age (categorical, 40–49, 50–59, 60–69 and ≥70 years), sex (categorical, male and female), household income (categorical, 1st, 2nd, 3rd and 4th quartiles), residence (categorical, capital city and metropolitan area), smoking behavior (categorical, never smoker, past smoker and current smoker), drinking behavior (categorial, none, 1-2 times per week, 3–4 times per week and ≥5 times per week) and Charlson Comorbidity Index (CCI), (continuous). Household income was derived from the insurance premium. The algorithm for calculating CCI using claims data was derived from elsewhere [20]. These covariates were adjusted at three different levels. Model 1 adjusted only for age, income and residence, while model 2 adjusted for smoking, drinking and physical activities in addition to model 1. In model 3, CCI was also considered.

The stratified analyses were performed for lifestyle behaviors—smoking, drinking and physical activity. Fully adjusted model 3 was used to determine the effects of each lifestyle behavior on the incidence of MetS. Multivariate logistic regression was also used to calculate the aORs with 95% CIs.

Statistical significance was defined as a *p*-value of <0.05 in a two-tailed manner. All data collection and statistical analyses were conducted using STATA 15.0 (StataCorp, College Station, TX, USA).

### 2.4. Ethics Committee Approval

This study was approved by the Seoul National University Hospital Institutional Review Board (IRB number: E-1806-076-951). The requirement for informed consent was waived as the NHIS-NSC database was anonymized according to strict confidentiality guidelines prior to distribution.

## 3. Results

Table 1 depicts the descriptive characteristics of the study population. The ranges of social trust for each quintile are 42–53%, 54–59%, 59–61%, 61–68% and 69–88%, respectively. There was no significant difference among the groups, except for the location of residence. There were no capital city dwellers in the 5th quintile of social trust group.

The aORs for MetS for the total population and male and female subgroups are shown in Table 2. A lower aOR for the incidence of MetS was shown in the 5th quintile group when compared to the 1st quintile of social trust in total and both sexes. Compared to the 1st quintile of social trust of the total population, the 2nd quintile group has an adjusted odds ratio of 0.88 (95% CI 0.80–0.96), the 3rd quintile 0.97 (0.88–1.07), the 4th quintile 0.87 (0.79–0.95) and the 5th quintile 0.87 (0.78–0.97) in model 1. The numbers did not differ significantly in models 2 and 3. In the case of males, the adjusted odds ratios with 95% CIs were 0.95 (0.83–1.05) in the 2nd quintile group, 0.96 (0.85–1.08) in the 3rd quintile group, 0.89 (0.79–1.01) in the 4th quintile and 0.88 (0.76–1.01) in the 5th quintile in all three models. The female population exhibited aORs with 95% CIs of 0.79 (0.66–0.95), 0.97 (0.82–1.15), 0.80 (0.66, 0.96) and 0.82 (0.68–0.98) in the 2nd, 3rd, 4th and 5th quintile groups, respectively, when compared to the 1st quintile group of social trust.

Table 3 shows the aORs of each MetS component when adjusted for age, residence, income, smoking, drinking, physical activity and CCI. Among the components of MetS, only waist circumference reduced the aOR of new incidences MetS in a statistically significant manner. The aOR for HDL-C was statistically significant only in the 5th quintile group of social trust. The aOR with 95% CI for waist circumference for the 5th quintile group of social trust when compared to the 1st quintile group was 0.92 (0.85–0.99) for model 1. The aORs with 95% CIs for model 2 and model 3 in the 5th quintile group were 0.91 (0.84–0.99) and 0.91 (0.84–0.99). When the 5th quintile group of social trust was compared to the 1st quintile group, the aOR with 95% CI was 0.88 (0.81–0.96) in model 1. For models 2 and 3, the aORs with 95% CIs were 0.88 (0.80–0.96) and 0.88 (0.80–0.95), respectively.

Lastly, stratified analyses on the association between social trust and MetS, taking into consideration smoking, alcohol intake and physical activity, are shown in Table 4. Never smokers and non-drinkers reduced the incidence of MetS. The aORs with 95% CIs in never smokers when compared to the 1st quintile group of social trust were 0.85 (0.75–0.97), 0.95 (0.84–1.09), 0.84 (0.74–0.95) and 0.85 (0.74–0.99) in the 2nd, 3rd, 4th and 5th quintiles groups. On the other hand, the aORs with 95% CIs for past and current smokers were 0.93 (0.81–1.06), 1.00 (0.87–1.16), 0.91 (0.79–1.05) and 0.89 (0.76–1.04) in the 2nd, 3rd, 4th and 5th quintile groups of social trust when compared to the 1st quintile population. In non-alcohol drinkers, the aORs when compared to the 1st quintile group of social trust were 0.89 (0.77–1.02), 1.03 (0.90–1.18), 0.82 (0.71–0.94) and 0.82 (0.70–0.95) in the 2nd, 3rd, 4th, and 5th quintile groups of social trust. The alcohol drinkers showed aORs with 95% CIs of 0.88 (0.77–1.00), 0.92 (0.80–1.05), 0.92 (0.81–1.05) and 0.93 (0.80–1.07) from the 2nd to the 5th quintiles of social trust. Then the physically active group and inactive group were also compared, and the protective effect of physical activity on MetS was not significant. The physically inactive group had aORs with 95% CIs of 0.81 (0.71–0.95) and 0.88 (0.75–1.03) in the 4th and 5th quintiles of social trust groups, respectively, when compared to the 1st quintile, while the physically active group had aORs with 95% CIs of 0.91 (0.80–1.04) and 0.87 (0.76–1.00) in the 4th and 5th quintile groups.

## 4. Discussion

This population-based, longitudinal study examined the association between social trust and MetS. The beneficial effect of social trust on reducing the incidence of MetS persisted even after taking into account differences in age, income, area of residence, lifestyle behaviors—smoking, alcohol drinking and physical activity—and CCI. In stratified analyses with lifestyle behaviors, smoking, alcohol intake and physical activity all showed a statistically significant impact on MetS incidence in a previously healthy population. To our knowledge, this is the first longitudinal study to demonstrate that district-level trust was associated with a lower incidence of MetS in individuals, using nationally representative cohort data.

Previous studies have investigated the association between social trust and proxy measures of MetS. In a Canadian paper that investigated the causes of health inequality, Indigenous people with higher social support were associated with a lower cardiovascular disease risk score [21]. Another study conducted with Americans aged 50 years and older, found a statistically significant association between higher perceived social cohesion and a lower incidence of stroke [22]. On the contrary, a nationally representative study conducted in China in 2017 showed that higher social trust was associated with a lower likelihood of obesity, and harmonious social relationships were correlated with higher chances of becoming obese [3]. Most of these investigations were cross-sectional and could not prove causality, while this study was longitudinally designed to capture the effect of social trust on the incidence of MetS. We also used the direct measure of MetS and its components rather than proxy measures. It was determined that higher social trust was associated with a lower incidence of MetS. Furthermore, by adjusting out age, income, area of residence, lifestyle behaviors and CCI, we tried to eliminate the confounding factors that were not pre-determined in the study design. The aORs of developing MetS remained lower in higher social trust groups even after adjusting for covariates.

Different mechanisms have been proposed to explain the association between social trust and MetS. First, people with higher social trust are likely to have a higher sense of security, which may help in the exchange of valuable information or instrumental support within society and in absorbing health-promoting behaviors [17,23]. Second, in societies with higher social support and network groups, people have easier access to transportation systems and healthcare [24]. Furthermore, when residents live in a safer neighborhood, they are more likely to exercise [12]. Another explanation is collective efficacy. Members of a community may act together to promote health-promoting behaviors and against harmful behaviors, such as collecting signatures for a smoking-free zone [25]. Lastly, psycho-social pathways also help explain the association between social trust and MetS. A lower level of social trust may increase social anxiety and stress, which in turn may elevate blood cortisol levels. The stimulation of the hypothalamic-pituitary-adrenal (HPA) axis can cause inflammation and diseases, such as cancer [26] and cardiovascular accidents [27].

In this study, the aORs of MetS incidence were found to be lower in women than in men. This may be explained by women having more a trusting and pro-social nature than men. Women’s tendency to adopt communal and interpersonal facilitative behavior may work together towards healthful behavior in a community [28]. In addition, women tend to relay information among members of a community more frequently than men. Men rely more on the information communicated with their spouses than with other community members. Moreover, the aORs were statistically less significant in the middle quintile groups of social trust. Social trust may need to be at the extreme ends to exert influence on people’s lifestyle behaviors. Generally, higher social trust was associated with positive health outcomes that can be explained by the abovementioned mechanisms. However, the relationship did not prove to be as significant in fasting blood glucose, blood pressure or triglyceride levels. These three components of MetS are more closely related to eating habits, which this study did not consider. The members of a community may share similar diet patterns and different diets affect metabolic profiles differently [29].

Several limitations must be considered when interpreting the results of this study. Social trust was measured at one point in time, and changes were not considered. Additionally, the participants were only followed up for a short period of time because HDL-C levels were only collected in 2009. It may have been insufficient to determine the effects of social trust on the development of MetS. However, social trust is usually influenced by the environment, which does not change rapidly. Because social trust is closely knitted into the lives of community members, one year may have been enough to exert influence over the members’ health outcomes. In addition, although we adjusted for household income and area of residence, we could not fully take into account the effects of the neighborhood environment, education level and friends on health outcomes. The education level and diversity of friends were associated with chances of becoming obese in previous studies [30,31]. Lastly, we excluded the samples from rural areas due to population biases towards older adults and higher levels of social trust. This study tried to be more representative of the general population of the country. In rural areas, social trust is high and MetS incidence is low. Further analyses may be necessary to determine the influence of social trust on MetS in rural adults.

In conclusion, higher social trust decreased the likelihood of developing MetS. Quitting smoking, drinking in moderation and being physically active also reduced the risk. Therefore, it is important to create a community where healthy lifestyles are encouraged among members of society. Since it is known that reducing MetS requires collective effort as a society, public health policy should aim to create health-conducive environments by increasing social trust through building recreational facilities and creating community memberships.

## Figures and Tables

**Table 1 ijerph-17-05629-t001:** Descriptive characteristics of the study population.

	Social Trust (Quintiles)					*p*-Value
	1st (Lowest)	2nd	3rd	4th	5th (Highest)	
Range, %	42–53	54–59	59–61	61–68	69–88	
Number of people, *N* (%)	7761 (20.76)	7996 (21.39)	6873 (18.39)	8212 (21.97)	6535 (17.48)	
Age, years, *N* (%)						0.114
40–49	4078 (52.54)	4126 (51.60)	3544 (51.56)	4205 (51.21)	3400 (52.03)	
50–59	2644 (34.07)	2813 (35.18)	2322 (33.78)	2865 (34.89)	2223 (34.02)	
60–69	829 (10.68)	869 (10.87)	797 (11.60)	925 (11.26)	712 (10.90)	
≥70	210 (2.71)	188 (2.35)	210 (3.06)	217 (2.64)	200 (3.06)	
Sex, *N* (%)						0.002
Male	3782 (48.73)	3920 (49.02)	3485 (50.71)	4031 (49.09)	3368 (51.54)	
Female	3979 (51.27)	4076 (50.98)	3388 (49.29)	4181 (50.91)	3167 (48.46)	
Household income, *N* (%)						<0.001
1st quartile (lowest)	1081 (13.93)	1121 (14.02)	975 (14.19)	1147 (13.97)	924 (14.14)	
2nd quartile	1898 (24.46)	1714 (21.44)	1466 (21.33)	1756 (21.38)	1289 (19.72)	
3rd quartile	2298 (29.61)	2285 (28.58)	1812 (26.36)	2244 (27.33)	1899 (29.06)	
4th quartile (highest)	2484 (32.01)	2876 (35.97)	2620 (38.12)	3065 (37.32)	3065 (37.32)	
Location of residence, *N* (%)						<0.001
Capital city	4490 (51.82)	4789 (54.00)	4307 (56.25)	2843 (31.46)	0 (0.00)	
Metropolitan	4175 (48.18)	4080 (46.00)	3350 (43.75)	6194 (68.54)	7272 (100.00)	
Smoking, *N* (%)						0.610
Never smoker	5450 (61.63)	5540 (62.40)	4756 (62.11)	5612 (62.10)	4454 (61.25)	
Past smoker	1368 (15.79)	1353 (15.26)	1193 (15.58)	1459 (16.14)	1182 (16.25)	
Current smoker	1957 (22.59)	1976 (22.28)	1708 (22.31)	1966 (21.76)	1636 (22.50)	
Alcohol consumption, times per week, *N* (%)						0.099
None	4442 (51.26)	4566 (51.48)	4057 (52.98)	4764 (52.72)	3758 (51.68)	
1–2	2959 (34.15)	3034 (34.21)	2614 (34.14)	3041 (33.65)	2512 (34.54)	
3–4	938 (10.83)	918 (10.35)	708 (9.25)	882 (9.76)	741 (10.19)	
≥5	326 (3.76)	351 (3.96)	278 (3.63)	350 (3.87)	261 (3.59)	
Physical activity, intensity, MVPA, *N* (%)						0.002
Physically inactive	3911 (45.14)	3840 (43.30)	3351 (43.76)	3914 (43.31)	3114 (42.82)	
1–2 times MVPA per week	1835 (21.18)	1954 (22.03)	1607 (20.99)	1863 (20.62)	1460 (20.08)	
3–4 times MVPA per week	1298 (14.98)	1345 (15.17)	1223 (15.97)	1448 (16.02)	1180 (16.23)	
≥5 times MVPA per week	1621 (18.71)	1730 (19.51)	1476 (19.28)	1812 (20.05)	1518 (20.87)	
Charlson Comorbidity Index, *N* (%)						0.025
0	3095 (35.72)	3113 (35.10)	2736 (35.73)	3081 (34.09)	2519 (34.64)	
1	3177 (36.66)	3234 (36.46)	2793 (36.48)	3336 (36.91)	2629 (36.15)	
2	1542 (17.80)	1630 (18.38)	1359 (17.75)	1630 (18.04)	1379 (18.96)	
≥3	851 (9.82)	892 (10.06)	769 (10.04)	990 (10.95)	745 (10.24)	

*p*-value calculated with chi-squared test for categorical variables and ANOVA for continuous variables. Abbreviations: MVPA, moderate to vigorous physical activity; CCI, Charlson Comorbidity Index.

**Table 2 ijerph-17-05629-t002:** Adjusted odds ratio (95% confidence intervals) for metabolic syndrome based on the NCEP ATP III criteria by social trust quintiles.

		Social Trust (Quintiles)					
		1st (Lowest)	2nd	3rd	4th	5th (Highest)	*p* For Trend
Total, *N*		7761	7996	6873	8212	6535	
Events, *N* (%)		640 (8.25)	580 (7.25)	556 (8.09)	589 (7.17)	449 (6.87)	
aOR (95% CI)	Model 1	Reference	0.88 (0.80–0.96)	0.97 (0.88–1.07)	0.87 (0.79–0.95)	0.87 (0.78–0.97)	0.012
	Model 2	Reference	0.88 (0.80–0.97)	0.98 (0.89–1.08)	0.87 (0.79–0.96)	0.88 (0.79–0.98)	0.021
	Model 3	Reference	0.88 (0.80–0.98)	0.98 (0.89–1.08)	0.87 (0.79–0.96)	0.88 (0.79–0.98)	0.022
		**Male**					
Total, *N*		3782	3920	3485	4031	3368	
Events, *N* (%)		368 (9.73)	363 (9.26)	334 (9.58)	350 (8.68)	247 (7.33)	
aOR (95% CI)	Model 1	Reference	0.93 (0.83–1.05)	0.96 (0.85–1.08)	0.89 (0.79–1.01)	0.88 (0.76–1.01)	0.049
	Model 2	Reference	0.93 (0.83–1.05)	0.97 (0.85–1.09)	0.89 (0.79–1.01)	0.88 (0.77–1.01)	0.063
	Model 3	Reference	0.93 (0.83–1.05)	0.97 (0.85–1.10)	0.90 (0.79–1.01)	0.88 (0.77–1.01)	0.066
		**Female**					
Total, *N*		3979	3572	3892	3605	3743	
Events, *N* (%)		272 (6.84)	194 (5.43)	245 (6.29)	207 (5.74)	234 (6.25)	
aOR (95% CI)	Model 1	Reference	0.82 (0.70–0.95)	0.95 (0.82–1.10)	0.84 (0.72–0.98)	0.83 (0.72–0.97)	0.057
	Model 2	Reference	0.82 (0.70–0.95)	0.95 (0.82–1.10)	0.84 (0.72–0.98)	0.83 (0.72–0.98)	0.054
	Model 3	Reference	0.82 (0.71–0.96)	0.95 (0.82–1.10)	0.84 (0.72–0.98)	0.83 (0.72–0.97)	0.054

Criteria for metabolic syndrome was defined as meeting three or more of the following conditions, as suggested by NCEP ATP III: (1) Impaired Fasting Glucose (≥100 mg/dL), (2) Elevated WC (>90 cm for men and >85 cm for women), (3) High Blood Pressure (SBP: ≥130 mmHg and DBP: ≥85 mmHg), (4) High Triglycerides (≥150 mg/dL), (5) Reduced HDL-cholesterol (<40 mg/dL for men and <50 mg/dL for women). Data presented are *N* (%) and aOR (95% CI). Logistics Model 1: adjusted for age, income and residence. Logistics Model 2: adjusted for age, income, residence, smoking, drinking and physical activities. Logistics Model 3: adjusted for age, income, residence, smoking, drinking, physical activities and Charlson Comorbidity Index. Abbreviations: aOR, Adjusted Odds Ratio; CI, Confidence Interval; WC, Waist Circumference; SBP, Systolic Blood Pressure; DBP, Diastolic Blood Pressure; HDL, High Density Lipoprotein; NCEP ATP, National Cholesterol Education Program-Adult Treatment Panel III. Note: Each category of social trust quartile is compared to the 1st social trust quartile (reference).

**Table 3 ijerph-17-05629-t003:** Adjusted odds ratio of (95% confidence intervals) for each metabolic syndrome criterion by social trust quintiles.

		Social Trust (Quintiles)					
		1st (Lowest)	2nd	3rd	4th	5th (Highest)	*p* For Trend
Components of Metabolic Syndrome	Variable						
High Blood Pressure		(*N* = 7605)	(*N* = 7839)	(*N* = 6741)	(*N* = 8055)	(*N* = 6333)	
aOR (95% CI)	Events (%)	714 (9.39)	743 (9.48)	635 (9.42)	710 (8.81)	583 (9.21)	
	Model 1	Reference	0.99 (0.91–1.08)	0.96 (0.87–1.05)	0.92 (0.84–1.01)	1.05 (0.95–1.15)	0.823
	Model 2	Reference	0.99 (0.91–1.08)	0.97 (0.88–1.06)	0.93 (0.85–1.02)	1.05 (0.95–1.16)	0.990
	Model 3	Reference	0.99 (0.91–1.09)	0.97 (0.88–1.06)	0.93 (0.85–1.02)	1.05 (0.95–1.16)	0.971
Abdomen Obesity		(*N* = 6720)	(*N* = 7021)	(*N* = 6054)	(*N* = 7229)	(*N* = 5849)	
aOR (95% CI)	Events (%)	791 (11.77)	703 (10.01)	700 (11.56)	750 (10.37)	623 (10.65)	
	Model 1	Reference	0.86 (0.80–0.92)	0.93 (0.85–1.00)	0.87 (0.81–0.93)	0.92 (0.85–0.99)	0.028
	Model 2	Reference	0.86 (0.80–0.92)	0.93 (0.86–1.00)	0.86 (0.80–0.93)	0.91 (0.84–0.99)	0.023
	Model 3	Reference	0.86 (0.80–0.92)	0.93 (0.86–1.00)	0.86 (0.80-0.93)	0.91 (0.84–0.99)	0.023
Impaired Fasting Glucose		(*N* = 6407)	(*N* = 6574)	(*N* = 5696)	(*N* = 6641)	(*N* = 5287)	
aOR (95% CI)	Events (%)	1157 (18.06)	1240 (18.86)	1163 (20.42)	1273 (19.17)	1065 (20.14)	
	Model 1	Reference	0.99 (0.93–1.06)	1.13 (1.06–1.21)	0.99 (0.93–1.06)	0.98 (0.91–1.05)	0.831
	Model 2	Reference	0.99 (0.93–1.06)	1.15 (1.07–1.23)	1.00 (0.94–1.07)	0.98 (0.91–1.06)	0.916
	Model 3	Reference	0.99 (0.93–1.06)	1.15 (1.07–1.23)	1.00 (0.94-1.07)	0.98 (0.91–1.06)	0.911
High Triglyceride		(*N* = 6464)	(*N* = 6721)	(*N* = 5642)	(*N* = 6875)	(*N* = 5416)	
aOR (95% CI)	Events (%)	929 (14.37)	880 (13.09)	782 (13.86)	902 (13.12)	780 (14.40)	
	Model 1	Reference	0.91 (0.85–0.97)	1.01 (0.94–1.08)	0.91 (0.85–0.98)	0.98 (0.910–1.05)	0.510
	Model 2	Reference	0.91 (0.85–0.98)	1.02 (0.95–0.99)	0.92 (0.86–0.99)	0.99 (0.92–1.07)	0.859
	Model 3	Reference	0.91 (0.85–0.98)	1.02 (0.95–1.10)	0.93 (0.86-0.99)	0.99 (0.92–1.07)	0.884
Low HDL-C		(*N* = 7117)	(*N* = 7275)	(*N* = 6234)	(*N* = 7384)	(*N* = 6014)	
aOR (95% CI)	Events (%)	814 (11.44)	804 (11.05)	721 (11.57)	865 (11.71)	676 (11.24)	
	Model 1	Reference	0.95 (0.89–1.03)	0.99 (0.92–1.07)	1.00 (0.93–1.08)	0.88 (0.81–0.96)	0.080
	Model 2	Reference	0.95 (0.88–1.03)	0.98 (0.90–1.06)	0.99 (0.92–1.07)	0.88 (0.80–0.96)	0.056
	Model 3	Reference	0.95 (0.88–1.03)	0.98 (0.90–1.06)	0.99 (0.92–1.07)	0.88 (0.80–0.95)	0.054

Metabolic syndrome diagnostic criteria include (1) Impaired Fasting Glucose (≥100 mg/dL), (2) Elevated WC (>90 cm for men and >85 cm for women), (3) High Blood Pressure (SBP: ≥ 130 mmHg and DBP: ≥ 85 mmHg), (4) High Triglyceride (≥150 mg/dL), (5) Reduced HDL-cholesterol (<40 mg/dL for men and <50 mg/dL for women). Data presented are N (%) and aOR (95% CI). Logistics Model 1: adjusted for age, income and residence. Logistics. Model 2: adjusted for age, income, residence, smoking, drinking and physical activities. Logistics. Model 3: adjusted for age, income, residence, smoking, drinking, physical activities and Charlson Comorbidity Index. Abbreviations: aOR, Adjusted Odds Ratio; CI, Confidence Interval; WC, Waist Circumference; SBP, Systolic Blood Pressure; DBP, Diastolic Blood Pressure; HDL, High Density Lipoprotein. Note: Each category of social trust quartile is compared to the 1st social trust quartile (reference).

**Table 4 ijerph-17-05629-t004:** Stratified analyses on the association of social trust with metabolic syndrome, taking into consideration smoking, alcohol intake and physical activity.

	Adjusted Odds Ratio (95% Confidence Interval)				
	Social Trust (Quintiles)				
	1st (Lowest)	2nd	3rd	4th	5th (Highest)
Stratified analysis					
Smoking					
Never smokers	Reference	0.85 (0.75–0.97)	0.95 (0.84–1.09)	0.84 (0.74–0.95)	0.85 (0.74–0.99)
Past and current smokers	Reference	0.93 (0.81–1.06)	1.00 (0.87–1.16)	0.91 (0.79–1.05)	0.89 (0.76–1.04)
Alcohol intake					
No	Reference	0.89 (0.77–1.02)	1.03 (0.90–1.18)	0.82 (0.71–0.94)	0.82 (0.70–0.95)
Yes	Reference	0.88 (0.77–1.00)	0.92 (0.80–1.05)	0.92 (0.81–1.05)	0.93 (0.80–1.07)
Physical activity					
No	Reference	0.92 (0.80–1.05)	0.91 (0.79–1.05)	0.82 (0.71–0.95)	0.88 (0.75–1.03)
Yes	Reference	0.85 (0.75–0.97)	1.04 (0.91–1.18)	0.91 (0.80–1.04)	0.87 (0.76–1.00)

Fully adjusted model includes adjustments for age, residence, household income and Charlson Comorbidity Index. Adjusted odds ratios were calculated by multivariate logistic regression after adjustments for age, residence, household income and Charlson Comorbidity Index.

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
