# Peer review of "Association between Social Trust and Metabolic Syndrome in a Previously Healthy Population—A Longitudinal Cohort Study in South Korea"

_ijerph, 2020, doi:10.3390/ijerph17165629_

Round 1
Reviewer 1 Report
This article concerns a topic of interest both in the sociological and public health fields. It's broken down clearly enough, but a few observations are needed.
In my opinion, the gender issue could be better explained both in the introduction and discussion.
In the methods, considering the observation period indicated, it seems that the analysis is retrospective. It would be correct to specify it. The same observation period, as stated by the authors, is too short and could invalidate the real longitudinal character of the study.
The authors should do a general linguistic check and correct some minor errors.
Reviewer 2 Report
Comments
This is an interesting study to explore the relationship between social trust and metabolic syndrome. The whole manuscript is well written, the results are consistent with our healthy lifestyles. In my opinion, this manuscript can be accepted after minor revision.
Minor suggestions:
1) “p” should be italic and lowercase "p"
2) page 135~136, the authors said that “HDL-C and waist circumference reduced the aOR of new incidence MetS in a statistically significant manner”. However, in Table 3, the p values were > 0.05 in “Low HDL-C group”. Please explain.
3) In addition, I don’t understand why in Table 2, the total number is not the sum of the respective female and male in different quintiles?
Reviewer 3 Report
To analyze the association between social trust and metabolic syndrome, authors performed one population-based, longitudinal study. The study participants were extracted from the Korean National Health Insurance Service. Authors found that higher social trust was associated with lower incidence of metabolic syndrome. Authors suggested that building community with psychosocial support might be helpful in improving public health.
Overall, this study is very interesting. The cohort is large and representative. The statistical method is appropriate. I think this manuscript can be accepted in the current version.
Major Comments:
- In abstract, research background was described too simple.
- This study is focused on the association between social trust and metabolic syndrome. However, authors didn’t describe in detail what is the social trust, it is recommended that authors describe more in the introduction part.
